# Atomically dispersed nickel as coke-resistant active sites for methane dry reforming

Mohcin Akri[1,12], Shu Zhao[2,12], Xiaoyu Li[1,3], Ketao Zang[4], Adam F. Lee [5], Mark A. Isaacs[6], Wei Xi[4], Yuvaraj Gangarajula[1], Jun Luo[4], Yujing Ren[1], Yi-Tao Cui[7], Lei Li[8], Yang Su[1], Xiaoli Pan[1], Wu Wen[9], Yang Pan[9], Karen Wilson[5], Lin Li[1], Botao Qiao [1,10]*, Hirofumi Ishii[11], Yen-Fa Liao[11], Aiqin Wang[1], Xiaodong Wang[1] & Tao Zhang[1,3]*

Dry reforming of methane (DRM) is an attractive route to utilize $CO_2$ as a chemical feedstock with which to convert $CH_4$ into valuable syngas and simultaneously mitigate both greenhouse gases. Ni-based DRM catalysts are promising due to their high activity and low cost, but suffer from poor stability due to coke formation which has hindered their commercialization. Herein, we report that atomically dispersed Ni single atoms, stabilized by interaction with Ce-doped hydroxyapatite, are highly active and coke-resistant catalytic sites for DRM. Experimental and computational studies reveal that isolated Ni atoms are intrinsically coke-resistant due to their unique ability to only activate the first C-H bond in $CH_4$, thus avoiding methane deep decomposition into carbon. This discovery offers new opportunities to develop large-scale DRM processes using earth abundant catalysts.

[1] CAS Key Laboratory of Science and Technology on Applied Catalysis, Dalian Institute of Chemical Physics, Chinese Academy Sciences, 116023 Dalian, China. [2] Beijing Guyue New Materials Research Institute, Beijing University of Technology, 100124 Beijing, China. [3] University of Chinese Academy of Sciences, 100049 Beijing, China. [4] Center for Electron Microscopy and Tianjin Key Lab of Advanced Functional Porous Materials, Institute for New Energy Materials, School of Materials Science and Engineering, Tianjin University of Technology, 300384 Tianjin, China. [5] Applied Chemistry & Environmental Science, RMIT University, Melbourne 3000, Australia. [6] Department of Chemistry, University College London, London, UK. [7] Synchrotron Radiation Laboratory, Laser and Synchrotron Research Center (LASOR), The Institute for Solid State Physics, The University of Tokyo, 1-490-2 Kouto, Shingu-cho Tatsuno, Hyogo 679-5165, Japan. [8] Synchrotron Radiation Research Center, Hyogo Science and Technology Association, Hyogo 679-5165, Japan. [9] National Synchrotron Radiation Laboratory, University of Science and Technology of China, Hefei 230029, China. [10] Dalian National Laboratory for Clean Energy, 116023 Dalian, China. [11] National Synchrotron Radiation Research Center, Hsinchu 30076, Taiwan (ROC). [12] These authors contributed equally: Mochin Akri and Shu Zhao. *email: bqiao@dicp.ac.cn; taozhang@dicp.ac.cn

D ry reforming of methane (DRM) is the process of converting methane ($CH_4$) and carbon dioxide ($CO_2$) into synthesis gas (syngas, $H_2 + CO$)[1–3], an important building block for world scale industrial processes and energy conversion such as Fischer-Tropsch (FT), carbonylation, hydroformylation, and for the syntheses of fuels and high value-added chemicals[3–6]. $CO_2$ and $CH_4$ are the two most important atmospheric greenhouse gases (GHGs) responsible for anthropogenic climate change[7], but are also abundant and low-cost carbon sources;[8] the latter is regarded as a (relatively) clean energy source with which to realize a low-carbon economy[9]. Despite the emergence of DRM around 30 years ago[10], its potential to mitigate rising GHG emissions and provide clean(er) fossil fuel utilization has sparked renewed interest in associated catalytic technologies[2,11–16].

Almost 30 years' of intensive studies have identified Ni catalysts as the most promising candidate for DRM due to their low cost and high initial activity (comparable to noble metal catalysts)[1,2,12,17]. However, there are no commercial DRM processes to date due in large part to in situ catalyst deactivation by carbon deposition (coking) and/or sintering of Ni species[12,18]. Previous studies have shown that support acidity[17,19] and the dimensions of catalytically active components strongly influence carbon deposition[12,20]. Several studies have identified a threshold Ni nanoparticle size (2 nm[21] or 7–10 nm[22]) below which the carbon deposition significantly decreases. However, small Ni particles generally exhibit poor thermal stability and are prone to sintering under DRM reaction conditions, presenting an intractable problem.

Single-atom catalysts (SACs) comprising isolated, individual metal atoms dispersed on a solid support, have recently emerged as a new frontier in catalysis and materials science[23–29]. Although generally considered thermodynamically unstable, recent studies have revealed instances wherein atomically dispersed metals can actually be much more thermally stable and durable than their nanocatalyst counterparts[30–34]. Considering that Ni atoms in SACs are certainly below the threshold size to be coke-resistant in DRM, well-designed stable Ni SACs should be an ideal candidate for DRM provided that they are sufficiently active. Herein, we demonstrate that Ni atomically dispersed over hydroxyapatite (HAP) is highly active and completely coke-resistant during high temperature DRM. $CeO_x$ doping of HAP traps isolated Ni atoms, preventing on-stream sintering and deactivation and resulting in an active and stable high-performance DRM catalyst. This finding provides a new route for the development of coke-resistant DRM catalysts.

## Results

### Structural characterization of Ni single-atom catalyst.
HAP and Ce-substituted HAP with a Ce metal loading of 5 wt% (denoted as HAP-Ce) were prepared by a previously reported co-precipitation (CP) method[35]. Ni atoms were subsequently deposited over the HAP support with a nominal loading of 0.5 wt% (denoted as $0.5Ni_1/HAP$), and the HAP-Ce support with loadings of 0.5 wt%, 1 wt%, and 2 wt% (denoted as $0.5Ni_1/HAP$-Ce, $1Ni_1/HAP$-Ce, and $2Ni_1/HAP$-Ce), by strong electrostatic adsorption (SEA)[30,36,37]. For comparison, Ni nanocatalysts with 10 wt% loading were prepared by the same method over HAP and HAP-Ce supports, denoted as 10Ni/HAP and 10Ni/HAP-Ce, respectively. Synthesis details are provided in the Methods section.

Actual Ni metal loadings of the as-prepared catalysts were determined by inductively coupled plasma atomic emission spectroscopy (ICP-AES), and were similar to the nominal values (Supplementary Table 1) except for 10Ni/HAP-Ce which

contained 8.5 wt% Ni. Textural properties of Ce-doped HAP were similar to the undoped support (Supplementary Fig. 1a and Supplementary Table. 2). However, $CO_2$ temperature-programed desorption (TPD, Supplementary Fig. 1b) showed that ceria addition decreased the density of both weak and strong base sites, indicating that $CeO_x$ is less basic than HAP.

The X-ray diffraction (XRD) pattern of HAP-Ce evidenced no crystalline $CeO_2$ phases (Supplementary Fig. 2a), suggesting that cerium is either highly dispersed on the HAP external surface or doped into the lattice. Ni and NiO reflections were also absent from all unreduced samples (Supplementary Fig. 2b), suggesting that Ni/NiO were highly dispersed on both supports even following high temperature calcination (500 °C). For samples subsequently reduced under $H_2$ at 500 °C, only 10 wt% Ni samples show reflections characteristic of fcc nickel metal at 44.6° (Supplementary Fig. 2c, indicated by arrows). Corresponding Ni particle sizes were calculated according to the Scherrer equation as 17.0 and 10.6 nm for 10Ni/HAP and 10Ni/HAP-Ce, respectively, suggesting the introduction of Ce inhibited the sintering of Ni nanoparticles. For 0.5 wt% loadings, the absence of crystalline Ni and/or NiO phases may reflect the formation of highly dispersed Ni species, or instrumental detection limit due to the low Ni loading. All samples were therefore examined by (aberration corrected) high-angle annular dark-field scanning transmission electron microscopy (ac-HAADF-STEM) to identify the nature of supported Ni species. Neither Ni nanoparticles nor clusters were visible in relatively low magnification images for as-prepared $0.5Ni_1/HAP$ and $0.5Ni_1/HAP$-Ce samples (Supplementary Fig. 3a, b), whereas higher magnification images (Supplementary Fig. 3c, d) clearly revealed the exclusive presence of isolated Ni atoms, confirming $0.5Ni_1/HAP$ and $0.5Ni_1/HAP$-Ce as SACs before reduction. Following 500 °C $H_2$ reduction the $0.5Ni_1/HAP$-Ce sample still predominantly comprised atomically dispersed Ni (Fig. 1a and Supplementary Fig. 4), in addition to a small number (~7% frequency, Fig. 1d) of sub-1.5 nm Ni clusters. For $1Ni_1/HAP$-Ce and $2Ni_1/HAP$-Ce catalysts, low magnification images (Supplementary Figs. 5 and 6) showed the presence of a low density of ~1 nm nanoclusters, indicating that nickel predominantly existed as single atoms (Fig. 1b–d). In contrast, the as-prepared, 10 wt% samples comprised small (~1–2 nm) NiO clusters/nanoparticles over both supports (Supplementary Fig. 7a, b), which grew post-reduction to generate a distribution of small (1–3 nm) and large (5–10 nm) nanoparticles for 10Ni/HAP-Ce (Fig. 1c and Supplementary Fig. 7c) and a broad nanoparticle distribution (2–17 nm) for 10Ni/HAP (Supplementary Fig. 7d) in accordance with XRD.

The coordination environment of Ni atoms was further determined by quasi in situ X-ray absorption spectroscopy (XAS) measurements. Ni K-edge X-ray near-edge absorption spectra (XANES) collected after 500 °C reduction show that $2Ni_1/HAP$-Ce possessed an intense white line reminiscent of NiO (Fig. 2a), whereas 10Ni/HAP-Ce exhibited a weaker white line and strong pre-edge feature more closely resembling Ni foil, characteristic of oxidized and metallic chemical states respectively. Corresponding extended X-ray absorption fine structure (EXAFS) spectra confirmed the dominant presence of Ni-Ni scattering distances at ~2.50 Å (Fig. 2b) for 10Ni/HAP-Ce. In contrast, $2Ni_1/HAP$-Ce exhibited strong Ni–O scattering at 2.05 Å, a significant 2.60 Å contribution attributed to Ni–Ce scattering, and only a weak Ni–Ni contribution (corresponding to a coordination number ~0.7); these observations are consistent with atomically dispersed Ni (Supplementary Table 3 and Supplementary Fig. 8). XAS data are thus in good agreement with AC-HAADF-STEM imaging (Fig. 1b).

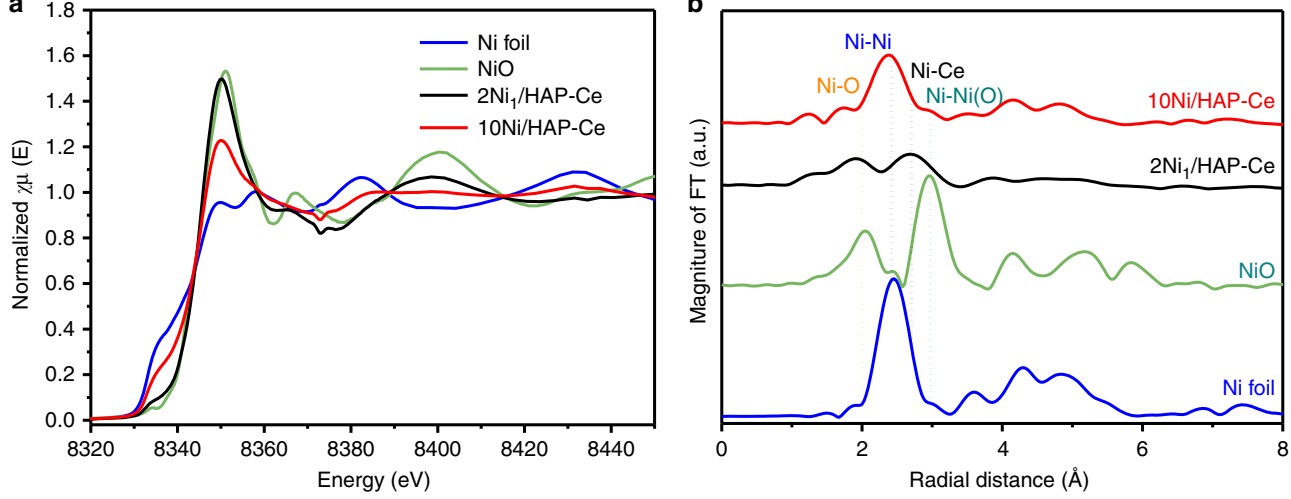

**Fig. 1** Electron microscopy images and size distribution of Ni/HAP-Ce samples. **a** HAADF-STEM images of 0.5Ni$_1$/HAP-Ce, **b** 2Ni$_1$/HAP-Ce, and **c** 10Ni/HAP-Ce samples after 500 °C H$_2$ reduction. **d** particle size distributions of **a**–**c**; yellow circles indicate atomically dispersed Ni and red squares indicate Ni metals nanoparticles. **e** EDX element maps of 0.5Ni$_1$/HAP-Ce

**Fig. 2** X-ray adsorption spectroscopy study of Ni/HAP-Ce catalysts. **a** Ni K-edge XANES spectra of 500 °C reduced HAP-Ce supported Ni catalysts and reference samples, and **b** corresponding phase shift corrected $k^2$-weighted Fourier transform

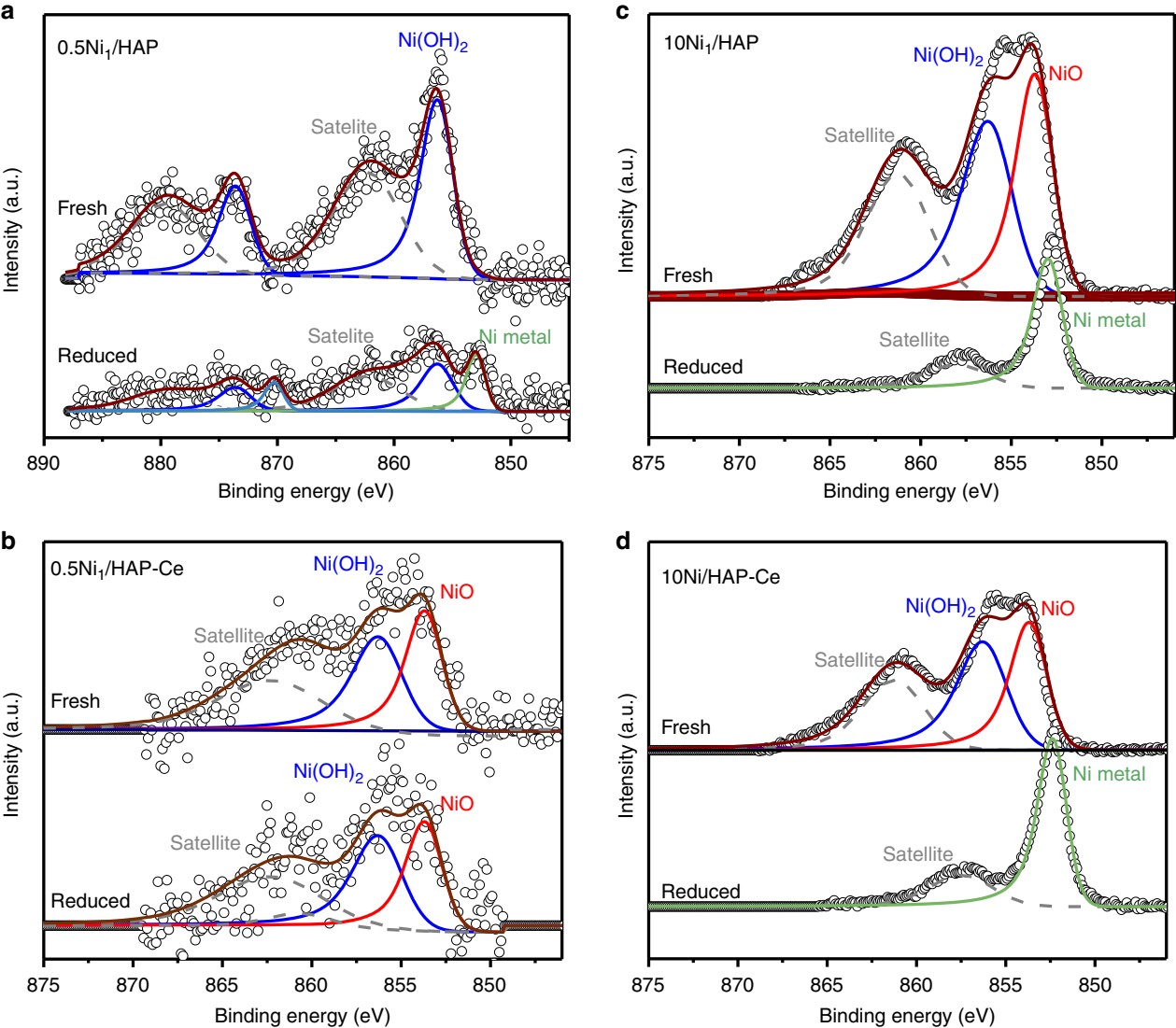

**Fig. 3** In-Situ XPS analysis of Ni/HAP and Ni/HAP-Ce samples. **a** High resolution Ni 2p XP spectra of as-prepared and in situ reduced: 0.5Ni$_1$/HAP, **b** 0.5Ni$_1$/HAP-Ce, **c** 10Ni/HAP, and **d** 10Ni/HAP-Ce

**Ni SAC electronic structure**. The surface oxidation state of Ni in the preceding as-prepared and reduced samples were studied by X-ray photoelectron spectroscopy (XPS). Since reduced Ni species are prone to ambient oxidation, 500 °C H$_2$ reduction was performed in situ within a quartz reaction cell prior to direct transfer into the spectrometer without air-exposure. As-prepared 0.5Ni$_1$/HAP sample contained only Ni(OH)$_2$ species at 856.3 eV binding energy (Fig. 3a), consistent with isolated Ni atoms at the hydroxyapatite interface, whereas the as-prepared 10Ni/HAP sample possessed almost equal amounts of Ni(OH)$_2$ and NiO (853.7 eV), Fig. 3c. In situ reduction of the 10Ni/HAP sample transformed all surface Ni$^{2+}$ species to metallic Ni (853.0 eV, Fig. 3c), indicating a relatively weak interaction with the HAP support, and in accordance with the observation of metallic Ni nanocluster/particles by STEM (Supplementary Fig. 7). In contrast, only half of the initial Ni(OH)$_2$ species in 0.5Ni$_1$/HAP underwent reduction to Ni metal (Fig. 3a), suggesting that isolated Ni adatoms experience a stronger support interaction. Figure 3d reveals that Ce doping had negligible impact on the Ni surface chemical state for high loadings, with equimolar Ni(OH)$_2$ and NiO species present in the as-prepared material, and only metallic Ni observed following in situ reduction. In contrast, the 0.5Ni$_1$/HAP-Ce differed significantly from its undoped counterpart, exhibiting both Ni(OH)$_x$ and NiO in an almost equimolar ratio in the as-prepared sample, and being entirely resistant to in situ reduction (Fig. 3b). Note that the high binding energy Ni 2p peak in Fig. 3b at 856.3 eV can be assigned to isolated Ni atoms in a distinct Ni(OH)$_x$ chemical environment, and not from Ni–O–Ni interactions in NiO nanoparticles. Non-local screening of photoexcited core holes only gives rise to peak splitting for the latter when NiO nanostructures exceed ~2.4 nm[38,39], of which there is no evidence in 0.5Ni$_1$/HAP-Ce by either HRTEM (Fig. 1a and Supplementary Figs. 3 and 4) or XRD (Supplementary Fig. 2). The surprising resistance 0.5Ni$_1$/HAP-Ce to H$_2$ reduction was confirmed by temperature-programmed reduction (TPR, Supplementary Fig. 9): high loading 10Ni/HAP and 10Ni/HAP-Ce exhibit almost complete reduction <600 °C associated with Ni metal formation (Supplementary Table. 4); in contrast, 0.5Ni$_1$/HAP sample exhibited a weak, broad reduction associated with loss of only ~60% of the initial Ni$^{2+}$ species, and 0.5Ni$_1$/HAP-Ce showed negligible reducibility. XPS and TPR unambiguously demonstrate that cerium doping of HAP stabilizes atomically dispersed Ni against reduction and sintering.

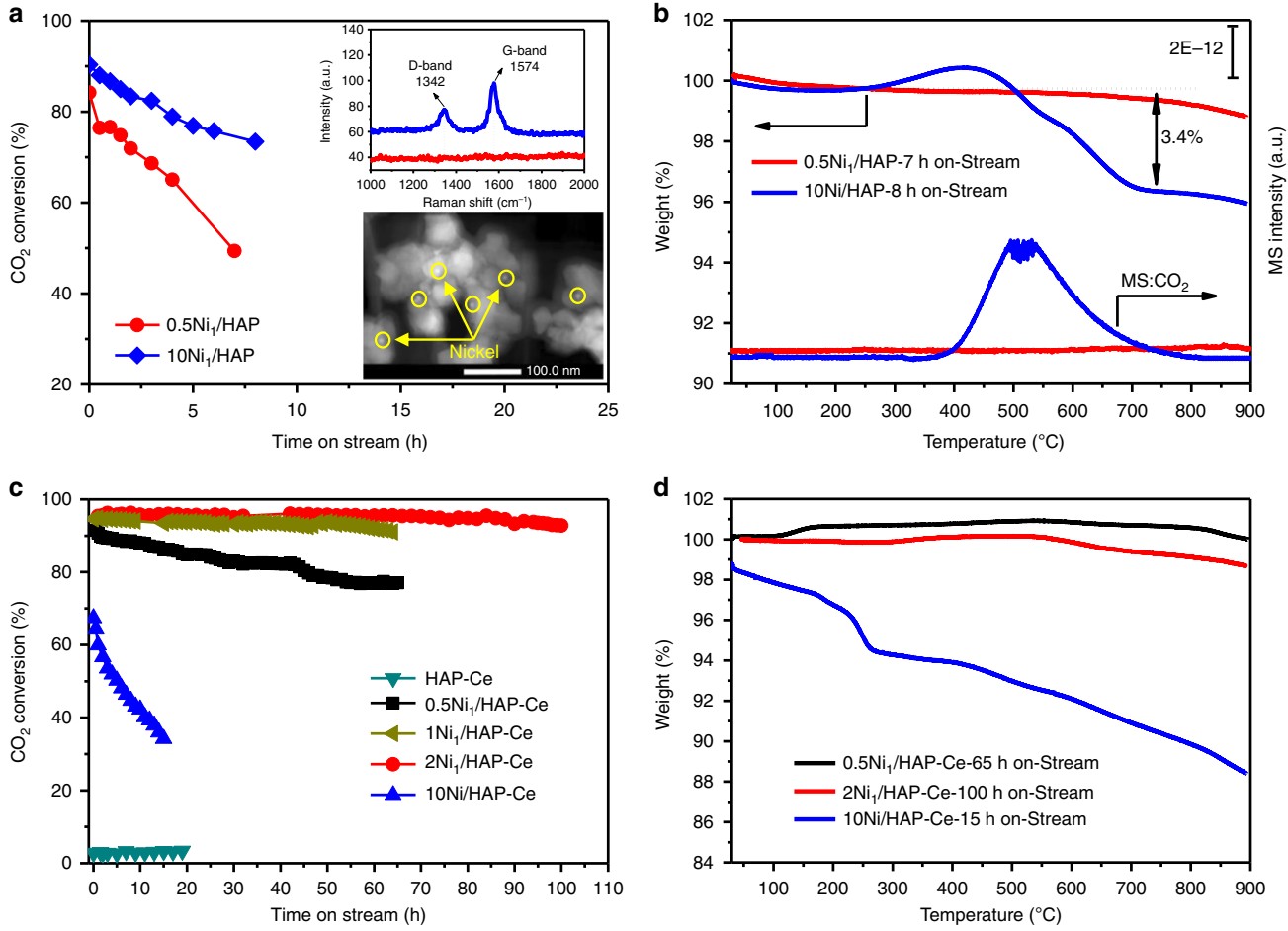

**Fig. 4** DRM performance and carbon deposition analysis over Ni/HAP and Ni/HAP-Ce samples. **a** $CO_2$ conversion during DRM over HAP and **c** HAP-Ce supported different Ni catalysts. Conditions: $T = 750\,°C$, $CH_4/CO_2/He = 10/10/30$, total flow = 50 mL min$^{-1}$ (GHSV = 60,000 mL h$^{-1}$ g$_{cat}$$^{-1}$), inset is Raman spectra of spent $0.5Ni_1/HAP$ and $10Ni/HAP$ after reaction at 750 °C and STEM image of $0.5Ni_1/HAP$ after 7 h reaction; **b–d** TGA-MS and TGA profiles of various catalysts after different reaction times on-stream at 750 °C

**Catalytic performance in DRM**. The preceding Ni catalysts were evaluated for DRM using a 20 vol% $CH_4$ + 20 vol% $CO_2$ feed gas (balanced with He) with a gas hourly space velocity (GHSV) of 60,000 mL g$_{cat}$$^{-1}$ h$^{-1}$. All samples were in situ reduced at 500 °C by 10 vol% $H_2$/He for 1 h prior to catalytic testing. At 750 °C, both $0.5Ni_1/HAP$ and $10Ni/HAP$ samples underwent rapid deactivation, with $CO_2$ conversions decreasing from 84 to 49% and 90 to 73% respectively over 7 h on-stream (Fig. 4a), mirroring corresponding $CH_4$ conversions (Supplementary Fig. 10a). However, characterization of post-reaction catalysts indicates that they deactivate by different mechanisms. Temperature-programmed oxidation revealed significant carbon deposition over the deactivated $10Ni/HAP$ (Fig. 4b), evidenced by a large weight loss associated with reactively formed $CO_2$ desorption between 400 and 600 °C. The nature of carbon species was analyzed by Raman spectroscopy (Fig. 4a inset). The peak at ~1328 cm$^{-1}$ (D band) corresponds to sp$^3$ carbon atoms at defects and disordered sites[40], while that ~1585 cm$^{-1}$ (G band) corresponds to sp$^2$ carbon atoms in graphitic rings[41]. In contrast, the $0.5Ni_1/$HAP sample exhibited negligible carbon deposition (Fig. 4b), albeit large nanoparticles (Fig. 4a inset) emerged from sintering of isolated Ni atoms, apparent from a 44.55° reflection in the post-reaction XRD pattern characteristic of fcc Ni metal (Supplementary Fig. 2d). It is important to note that the $0.5Ni_1/HAP$ SAC shows comparable initial activity to the $10Ni/HAP$ despite possessing twenty times less active component. HAP supported

isolated Ni atoms are therefore highly active and resistant to coking, albeit susceptible to sintering during high temperature DRM.

HAADF-STEM (Fig. 1a, b), $H_2$-TPR (Supplementary Fig. 9) and in situ XPS (Fig. 3b) demonstrate that cerium doping of the HAP support stabilizes atomically dispersed Ni to high temperature reduction. This results in a striking performance enhancement for the $0.5Ni_1/HAP$-Ce catalyst, for which $CO_2$ (Fig. 4c) and $CH_4$ (Supplementary Fig. 10b) conversions show only a modest decrease from initial ~90% levels over 65 h reaction. Increasing the Ni loading over the Ce-doped support to 1 wt% and 2 wt% further improved the long-term stability, with such catalysts of $1Ni_1/HAP$-Ce and $2Ni_1/HAP$-Ce (mainly composed of single atoms as revealed by Fig. 1b–d, Supplementary Figs. 5 and 6) able to continuously operate at 95% $CO_2$ conversion for almost 65 and 100 h, respectively, with a $H_2/CO$ ratio around unity (Supplementary Fig. 10c). These $Ni_1/HAP$-Ce SACs showed negligible coking (Fig. 4d) and theoretical carbon balance equal to 100% (Supplementary Fig. 10c). By comparison, the $10Ni/HAP$-Ce rapidly deactivated within 15 h on-stream (Fig. 4c) due to severe coking (Fig. 4d) along with a decrease in carbon balance and $H_2/CO$ ratio (Supplementary Fig. 10c). The principal role of Ce therefore appears to be to stabilize atomically dispersed Ni, rather than to suppress carbon deposition. In addition, a control experimental shows that HAP-Ce itself is almost inactive for DRM (Fig. 4c), suggesting that activity

| Table 1 Reaction rate of Ni/HAP and Ni/HAP-Ce catalysts | | | | | | | |
| --- | --- | --- | --- | --- | --- | --- | --- |
| Catalysts | Ni loading (wt%) | Specific rate (mol. $g_{Ni}^{-1}$ $h^{-1}$) | | TOF ($s^{-1}$) | | Temperature (°C) | Note |
| | | $CH_4$ | $CO_2$ | $CH_4$ | $CO_2$ | | |
| $0.5Ni_1$/HAP-Ce | 0.5 | 373.1 | 729.4 | 6.1 | 11.9 | 750 | This work |
| $2Ni_1$/HAP-Ce | 2.6 | 196.4 | 330.1 | 5.8 | 9.8 | 750 | This work |
| $10Ni$/HAP-Ce | 8.5 | 88.1 | 132.4 | 6.5 | 9.8 | 750 | This work |
| $Ni@SiO_2$ | 3.6 | 1.8 | – | 0.1 | N.A | 550 | Ref. [58] |
| $Ni/SiO_2@SiO_2$ | 0.16 | 144 | 289 | 21.2 | 42.5 | 800 | Ref. [20] |
| $Ni/La_2O_3$-LOC | 5.7 | 8.3 | – | 7.6 | N.A | 700 | Ref. [59] |
| $Ni-Zr/SiO_2$ | 8.3 | 0.25 | 0.35 | 1.06 | 1.48 | 450 | Ref. [60] |
| $Ni-Si/ZrO_2$ | 7.8 | 0.49 | 0.46 | 1.38 | 1.3 | 450 | Ref. [60] |

predominantly originates from the atomically dispersed Ni. It could be argued that the catalyst reduction temperature is quite low compared with the DRM reaction temperature, even though $H_2$-TPR result suggests that our Ni SAC would not be significantly reduced even at 750 °C. To confirm the sinter resistance of our catalysts, the DRM performance of $0.5Ni_1$/HAP-Ce and $2Ni_1$/HAP-Ce pre-reduced at 750 °C was also examined (Supplementary Fig. 11); there was no deterioration relative to 500 °C pre-reduced catalysts.

Recent elegant work by Tang et al.[42] reported a synergy between single-atom $Ru_1$ and $Ni_1$ sites resulting in very high activity for DRM even at 600 °C. Significant performance benefits accrued for our catalytic system under identical reaction conditions, even for loadings that do not exclusively give rise to single atoms: the 2 wt% $Ni_1$/HAP-Ce catalyst delivered similar activity and superior stability (Supplementary Fig. 11c) without recourse to precious metals.

The faster deactivation of $0.5Ni_1$/HAP-Ce versus $2Ni_1$/HAP-Ce merits consideration. In theory, all active sites within single-atom catalysts are identical and homogenously distributed. However, in practice the fabrication of a completely homogenous dispersion of single-atom sites is extremely difficult, and there is likely a distribution of coordination numbers and micro-environments[43]. For $0.5Ni_1$/HAP-Ce, we suggest that some single atoms initially present in lower coordination environments (e.g. the edges of HAP crystals) aggregate during reaction to lower their surface energies, resulting in the (albeit very slow) observed deactivation (Fig. 4c and Supplementary Fig. 10b). In contrast, unstable Ni single atoms present in the as-prepared $2Ni_1$/HAP-Ce likely sintered during subsequent reduction to form the small clusters and nanoparticles seen in Fig. 1b. Hence the reduced $2Ni_1$/HAP-Ce catalyst may possess a smaller proportion of unstable single atoms than the reduced $0.5Ni_1$/HAP-Ce catalyst. Note that nanoclusters and nanoparticles also eventually sinter during reaction, but with slower kinetics than isolated atoms (reflecting the higher surface energy and hence instability of the latter). Moreover, even if all isolated Ni atoms in $0.5Ni_1$/HAP-Ce initially occupy low energy surface sites, provided that their interaction with the support is weaker than the cohesive energy between Ni atoms then thermodynamics will always drive sintering for a sufficiently high energy input, at high temperature.

The specific activity of our HAP-Ce supported Ni catalysts was subsequently measured under differential reactor operation at much higher GHSV (528,000–1,056,000 mL $h^{-1}$ $g_{cat}^{-1}$) and low $CH_4/CO_2$ conversions (≤25%) to eliminate possible mass- and heat-transfer limitations. Ni/HAP-Ce SACs exhibited much higher specific activity (2–4 times), but an almost identical turnover frequency (TOF), to the nickel nanoclusters and

nanoparticles present for $10Ni$/HAP-Ce. The latter suggests a common $Ni^{2+}$ active site (likely $Ni(OH)_2$), and the former more efficient Ni utilization in the SACs[30,44–46] (a common feature and key advantage of SACs, although less important for earth abundant metal catalysts). Alternatively, the similar TOFs may be a simple coincidence of common energy barriers for different rate-limiting steps over Ni single atoms and nanoparticles. $CH_4$ activation is generally regarded as rate-limiting for overall DRM[14,47], with a barrier of ~0.9 eV over Ni(111), representative of the surface of Ni nanoparticles in $2Ni_1$/HAP-Ce, and $10Ni$/HAP-Ce[14]. This barrier is very similar to that calculated for activation of the first C–H bond in $CH_3O$ dehydrogenation over Ni single atoms in $0.5Ni_1$/HAP-Ce (the proposed rate-limiting step, see below). The specific activity of the $Ni_1$/HAP-Ce SAC is comparable or superior to that of literature Ni DRM catalysts (Table 1).

**Origin of Ni SAC coke resistance.** DRM usually proceeds according to the following steps: $CH_4$ decomposes over Ni active sites to form carbon and $H_2$ (Supplementary Fig. 12) and the resulting carbon is then oxidized by O adatoms liberated by $CO_2$ dissociation either over the support or at the metal-support interface[13,14,48]. High temperature (>700 °C) coke deposits thus predominantly arise from unoxidized carbon bound to Ni. The coke resistance of Ni SACs may be either intrinsic (i.e. they do not produce carbon from methane) or extrinsic (carbon formed is immediately oxidized by $CO_2$); the latter implies superior $CO_2$ activation. To distinguish between these two mechanisms, carbon deposition in the presence of $CH_4$ alone was studied at the reaction temperature (750 °C) over $2Ni_1$/HAP-Ce and $10Ni$/HAP-Ce catalysts (Fig. 5a, b). In both cases $CH_4$ decomposition occurred, however the resulting products were different. For $10Ni$/HAP-Ce, significant coking was evidenced by a color change (Supplementary Fig. 13a, b) and the large weight loss (8.67%) following TPO post-reaction (Fig. 5c), accompanied by $H_2$ and ethylene formation (Fig. 5b). In contrast, $2Ni_1$/HAP-Ce produced almost no coke (Supplementary Fig. 13c, d, Fig. 5c), but hydrogen, ethylene. It should be noted that on $2Ni_1$/HAP-Ce much greater C–C coupling occurred, but the $H_2$ yield was much lower compared with $10Ni$/HAP-Ce (Fig. 5d). The Ni SAC therefore favors partial $CH_4$ dehydrogenation and C–C coupling versus complete decomposition to C (step (4) in Supplementary Fig. 12). This observation is in good agreement with Fe single site catalysts, wherein non-oxidative coupling of $CH_4$ to ethene and aromatics without attendant coke formation was recently reported[49]. Theoretical calculations also suggest that oxide supported Pt SAC catalysts only efficiently activate the first C–H bond in $CH_4$, consistent with our present observations[50].

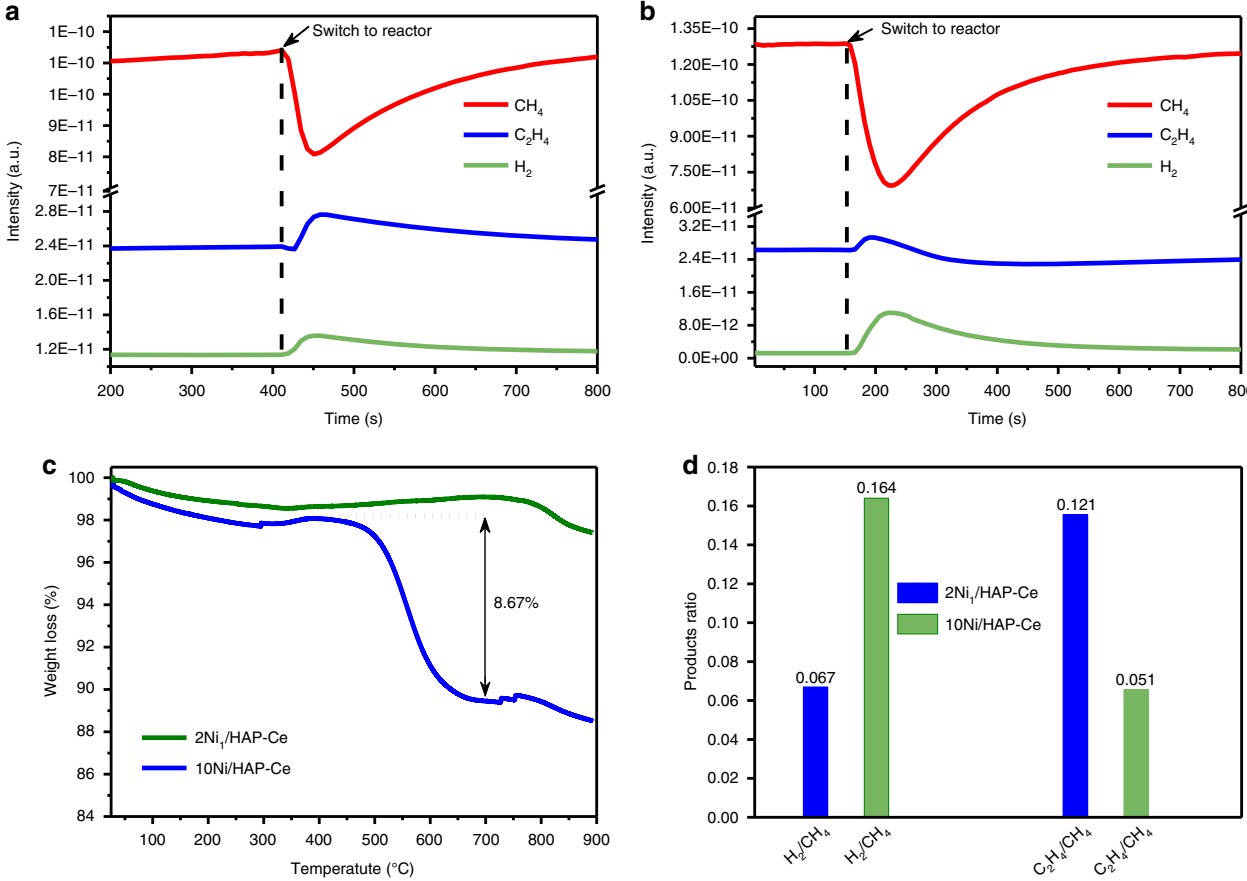

**Fig. 5** CH$_4$ decomposition and products distribution over Ni/HAP-Ce catalysts. **a** Mass spectrometer signals during CH$_4$ decomposition at 750 °C of 2Ni$_1$/HAP-Ce and **b** 10Ni/HAP-Ce, **c** TGA analysis of both catalysts after 1 h of reaction, and **d** product ratios for non-oxidative methane decomposition. H$_2$/CH$_4$ and C$_2$H$_4$/CH$_4$ represent the mass spectrometer product ratio of H$_2$ and C$_2$H$_4$ relative to the amount of CH$_4$ reacted

To understand the intrinsic coke resistance of our Ni SACs, underpinned by their selective activation of the first C–H bond, we undertook DFT calculations to investigate the first two steps of CH$_4$ dissociation over a Ni$_1$/CeO$_2$ surface (Fig. 6a) and subsequent transformation of CH$_3$ species (Fig. 6b, c). CH$_4$ adsorption on Ni$_1$/CeO$_2$ (111) is a physisorption process with a small adsorption energy of −0.11 eV, in which the carbon atom of CH$_4$ is located 2.56 Å above the Ni atom. The first C–H bond cleavage is exothermic by 0.65 eV with an activation barrier of 0.63 eV (**TS1**). This barrier is considerably lower than that over a Ni(111) surface (0.9 eV)[51], and hence activation of the first C–H bond in CH$_4$ is extremely efficient over ceria-bound isolated Ni atoms. The resulting CH$_3$ species binds atop the Ni atom while the H adatom moves to a neighboring O. The next dehydrogenation step, CH$_3$ → CH$_2$ + H, is thermodynamically unfavorable (endothermic by 0.80 eV) with a high activation barrier of 1.54 eV (**TS2**), and hence CH$_3$ dissociation is strongly disfavored over Ni$_1$/CeO$_2$, in excellent agreement with the preceding experiments. In light of the thermodynamic stability of CH$_3$ to subsequent C–H cleavage over our catalysts, the appearance of CO as a major product in our catalytic system appears paradoxical. However, a recent report of DRM over Ni$_4$ clusters (which also show good carbon deposition resistance[52]) identified the oxidation of CH$_3$ to CH$_3$O as an alternative route to CO following initial methane activation, as also proposed for Ni$_1$–Ru$_1$/CeO$_2$[42]. We therefore considered this alternative route for transforming reactively formed CH$_3$ to CO without carbon deposition (Fig. 6b, c). Unlike CH$_3$ dehydrogenation, the oxidation of CH$_3$ to CH$_3$O on Ni$_1$/CeO$_2$ is exothermic with a relatively lower barrier of 0.69 eV.

Subsequent CH$_3$O dehydrogenation to CHO is highly exothermic, and the rate-determining step (C–H bond activation in CH$_3$O proceeds with a barrier of 0.90 eV) to the final CO product (which is only slightly endothermic by 0.39 eV). The oxidation of CH$_3$ to CH$_3$O is hence plausible as the dominant (indirect) pathway for CH$_3$ dehydrogenation to CO. Overall, DFT calculations suggest that Ni$_1$/CeO$_2$ should be an active catalyst for CH$_4$ reforming to CO.

In summary, this work demonstrates that Ni atoms atomically dispersed over HAP are highly resistant to coking during DRM, but prone to on-stream deactivation through sintering. Ce doping of HAP induces strong metal-support interactions which stabilize Ni single atoms towards sintering, and favor selective activation of only the first C–H bond in methane, resulting in a high activity and stability for 100 h DRM with negligible carbon deposition. The excellent performance and stability of Ni single-atom catalysts offers a low-cost route to commercial dry methane reforming. Of broader significance, the findings in this work may provide a new approach for the development of highly coke-resistant Ni-based catalysts.

## Methods

**Chemicals.** Calcium nitrate (99%), cerium nitrate (99%), nickel nitrate (98.5%), ammonium dihydrogen phosphate (99%), and ammonia solution (25 vol% aqueous) were purchased from Sigma-Aldrich and used without pretreatment.

**Synthesis of hydroxyapatite and Ce-substituted HAP.** Stoichiometric hydroxyapatite (Ca:P = 1.67 molar ratio) and Ce substituted hydroxyapatite nanoparticles (5 wt% Ce loading), denoted as HAP and HAP-Ce respectively, were synthesized by co-precipitation of calcium nitrate (Ca(NO$_3$)$_2$.4H$_2$O) and

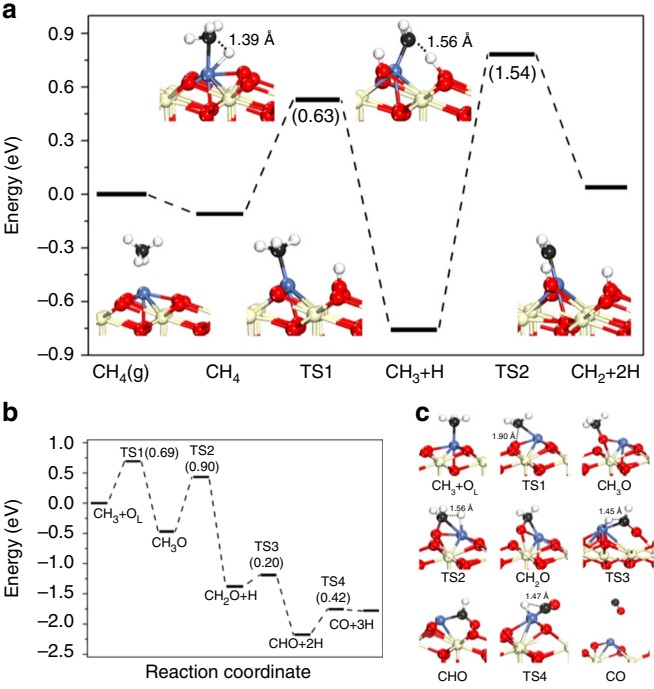

**Fig. 6** DFT calculation of $CH_4$ decomposition. **a** Potential energy diagram from DFT calculations of $CH_4$ dissociation over $Ni_1/CeO_2$. **b** Potential energy diagram from DFT calculations of $CH_3$ oxidation and $CH_3O$ dehydrogenation, and **c** corresponding geometries over $Ni_1/CeO_2$. Numbers in parentheses indicate the activation barriers for elementary steps in eV. Optimized structures for reaction intermediates are shown inset (Ce: yellow, Ni: blue, O: red, C: black, H: white)

ammonium dihydrogen phosphate $((NH_4)_2HPO_4)$, with or without cerium nitrate $(Ce(NO_3)_2.6H_2O)$. Typically, 7.54 g of $Ca(NO_3)_2.4H_2O$, and 2.54 g of $(NH_4)_2HPO_4$ were separately dissolved in 80 cm³ of doubly distilled water, and the pH adjusted to 10.5 with 25 vol% ammonia solution. Ammonium dihydrogen phosphate was added dropwise to the calcium nitrate solution at room temperature, and the resulting milky suspension stirred at 90 °C and 600 rpm for 2 h. The precipitate was rinsed with demineralized water three times prior to drying in air at 80 °C overnight. The dried sample was subsequently calcined at 400 °C (ramp rate 10 °C min⁻¹) in muffle furnace for 4 h. Ce-substituted hydroxyapatite was synthesized identically except for the addition of the appropriate amount of cerium nitrate to calcium nitrate prior to pH adjustment.

**Ni deposition on HAP and HAP-Ce.** A range of nominal Ni loadings (0.5, 1, 2, and 10 wt%) were deposited on the preceding supports by strong electrostatic adsorption at room temperature. An appropriate amount of nickel nitrate (Ni $(NO_3)_2$, $6H_2O$) was dissolved in 50 mL deionized water, and the solution pH adjusted to 10 with 25 vol% ammonia solution. The appropriate mass of HAP or HAP-Ce was then added to the nickel nitrate solution, and stirred at 600 rpm for 3 h at room temperature. The resulting impregnated solids were filtered, washed repeatedly with deionized water, and dried at 80 °C overnight prior to calcination at 500 °C (ramp rate 10 °C min⁻¹) for 4 h.

**Characterization.** Actual Ni loadings were determined by inductively coupled plasma spectrometry-atomic emission spectrometry (ICP-AES) on an IRIS Intrepid II XSP instrument (Thermo Electron Corporation). Specific surface areas ($S_{BET}$) were measured by $N_2$ porosimetry on a Micromeritics ASAP 2010 apparatus with adsorption-desorption isotherms recorded at 77 K. Samples were outgassed at 523 K prior to analysis. Surface areas were calculated from the adsorption branch using the Brunauer-Emmett-Teller method, with pore diameters determined from the desorption branch using the BJH method. X-ray diffraction (XRD) patterns were collected on a PW3040/60 × ' Pert PRO (PANalytical) diffractometer with Cu K$_\alpha$ radiation (0.15432 nm) operating at 40 kV and 40 mA. Patterns were collected between $2\theta = 10–80°$ at a scan speed of 10° min⁻¹. $CO_2$ temperature-programmed desorption (TPD) was performed on a Micromeritics AutoChem II 2920 instrument. HAP and HAP-Ce supports were first loaded into a U-shaped quartz reactor, pretreated at 300 °C in He for 0.5 h to remove physisorbed water and surface carbonates, and cooled to 50 °C. Base sites were subsequently saturated with $CO_2$ under flowing 10 vol% $CO_2$ in He at room temperature. Samples were then purged under He for 30 min prior to heating at 10 °C min⁻¹ to 900 °C under flowing He

and desorbing $CO_2$ detected by TCD. Temperature-programmed reduction ($H_2$-TPR) was also carried out on a Micromeritics AutoChem II 2920 instrument. Samples (100 mg) were placed in a U-shaped quartz reactor and pretreated in flowing Ar (30 mL min⁻¹) at 573 K for 30 min, and then ramped to 1073 K at 10 C min⁻¹ in flowing 10 vol% $H_2$/Ar (50 mL min⁻¹) and reacted $H_2$ detected by TCD. High-angle annular dark-field scanning transmission electron microscopy (HAADF-STEM) was performed on a JEOL JEM-2100F instrument. Aberration-corrected HAADF-STEM was performed on a JEOL JEM-ARM200F equipped with a CEOS probe corrector with a guaranteed resolution of 0.08 nm. Samples were dispersed by ultrasonication in ethanol, and the resulting solution dropped on to carbon films supported on copper grids.

In situ X-ray photoelectron spectroscopy was performed using a Kratos Axis Ultra-DLD fitted with an electron flood gun neutralizer and a monochromated aluminum X-ray source (1486.8 eV). As-prepared samples were analyzed under UHV conditions (<10⁻⁹ Torr) at 40 eV pass energy, and then transferred in vacuo to a quartz high-pressure reaction chamber where they were exposed to a 1 bar static atmosphere of 2 vol% $H_2$ in $N_2$ and heated to 500 °C at a ramp rate of 10 °C min⁻¹ for 1 h. Reduced samples were then cooled to room temperature and transferred in vacuo back to the analysis chamber for re-measurement. Spectra were energy referenced to adventitious carbon (284.8 eV), and Shirley background subtracted and fitted using CasaXPS v2.3.19PR1.0. Ni 2p XP spectra were fitted using a Doniach-Sunjic lineshape convoluted with a Gaussian-Lorentz function.

The X-ray absorption fine structure (XAFS) experiment was performed at the bending magnet beamline BL12B of SPring-8 (8 GeV, 100 mA) belong to National Synchrotron Radiation Research Center, in which the X-ray beam was monochromatized with water-cooled Si (111) double-crystal monochromator and focused with two Rh coated focusing mirrors with the beam size of 2.0 mm in the horizontal direction and 0.5 mm in the vertical direction around sample position. The samples were filled in a phi10 stainless tube for XAFS measurements. All of the samples were measured by both transmission and fluorescence modes at Ni K-edge. The samples were also measured at BL14W1 of the Shanghai Synchrotron Radiation Facility (SSRF) light sources and similar results were obtained. The spectra were analyzed and fitted using an analysis program Demeter[53]. For EXAFS fittings, the crystal structures of Ni foil and NiO are from Materials Project (https://materialsproject.org). For $2Ni_1/HAP-Ce$ sample the structure from DFT calculation is used.

**DFT calculations.** All calculations were performed using periodic DFT methods as implemented in the Vienna ab-initio simulation package (VASP)[54,55]. The projector augmented wave (PAW) method was used for the interaction between atomic cores and valence electrons. Valence orbitals of Ce (4f, 5s, 6s, 5p, 5d), Ni (3d, 4s), O (2s, 2p), C (2s, 2p), and H (1s) were described by plane-wave basis sets with cutoff energies of 400 eV, whereas the Brillouin zone was sampled at the Γ-point. Exchange-correlation energies were calculated via the generalized gradient approximation (GGA) with the PBE functional[56]. Spin-polarized DFT + U calculations with a value of $U_{eff} = 5.0$ eV for the Ce 4f state were applied to correct the strong electron-correlation properties of $CeO_2$. Convergence criteria for the electronic self-consistent iteration and force were set to 10⁻⁴ eV and 0.03 eV/Å respectively. For evaluating the energy barriers, transition states and pathways were computed using the climbing image nudged elastic band (CI-NEB) method[57]. The $CeO_2$ (111) surface was modeled by $p(3 \times 3)$ 9 atomic layer supercells with the bottom three layers fixed. $Ni_1/CeO_2$ was established by supporting one Ni atom on a $CeO_2$ (111) surface (Supplementary Fig. 14). The vacuum gap was set as ~15 Å to avoid the interaction between periodic images. Reaction energies and barriers were calculated by $E_r = E(FS) — E(IS)$ and $E_a = E(TS) — E(IS)$, where $E(IS)$, $E(FS)$ and $E$ (TS) are the energies of the corresponding initial state (IS), final state (FS), and transition state (TS), respectively.

**Catalytic testing.** Catalytic activity tests were performed in a fixed-bed quartz reactor (5 mm i.d.) under atmospheric pressure. Prior to each reaction, 50 mg of catalyst was placed in the reactor tube between quartz wool plugs and reduced in situ at 500 °C or 750 °C under flowing 10 vol% $H_2$/He (60 mL min⁻¹) for 1 h, and then purged with He for 30 min. Dry reforming was then undertaken using a mixture of 20 vol% $CO_2$ + 20 vol% $CH_4$ in He (50 mL min⁻¹ total flow rate) to generate a gas hourly space velocity (GHSV) of 60,000 mL $g_{cat}^{-1}$ h⁻¹. Specific activities for $CH_4$ and $CO_2$ conversion were measured at much higher GHSV (using 2.5–5 mg of catalyst, wherein the $CH_4$ and $CO_2$ conversions were <25%) during the first 60 min of reaction. Reactant and product concentrations were analyzed by an Agilent 6890 online gas chromatograph equipped with a TDX-01 column and a thermal conductivity detector, using He as a carrier gas. $CH_4$ and $CO_2$ conversions were calculated respectively according to the following Eqs. 1 and 2:

$$C_{CH_4} = \frac{F^{inlet} \times [CH_4] - F^{outlet} \times [CH_4]^{outlet}}{F^{inlet} \times [CH_4]^{inlet}} \quad (1)$$

$$C_{CO_2} = \frac{F^{inlet} \times [CO_2] - F^{outlet} \times [CO_2]^{outlet}}{F^{inlet} \times [CO_2]^{inlet}} \quad (2)$$

Where $[CH_4]$ and $[CO_2]$ are the respective mole fractions of $CH_4$ and $CO_2$ in the

feedstream, and $F^{inlet}$ and $F^{outlet}$ represents the total gas flow rate (ml min$^{-1}$) of inlet and outlet, respectively. Specific activities for $CH_4$ and $CO_2$ (mol $g_{Ni}^{-1}$ h$^{-1}$) were calculated from Eqs. 3 and 4:

$$\text{Rate CH}_4 = \frac{nCH_4}{\text{Weight of catalyst} \times W_{Ni}} \times 60 \qquad (3)$$

$$\text{Rate CO}_2 = \frac{nCO_2}{\text{Weight of catalyst} \times W_{Ni}} \times 60 \qquad (4)$$

where,

$$nCH_4 = \frac{\left(F^{inlet} \times [CH_4] \times C_{CH_4}\right)}{(22.4 \times 10^3)} \text{ (mol)}$$

$$nCO_2 = \frac{\left(F^{inlet} \times [CO_2] \times C_{CO_2}\right)}{(22.4 \times 10^3)} \text{ (mol)}$$

and $F^{inlet}$ represents the total flow (ml min$^{-1}$),$[CH_4]$, $[CO_2]$, $C_{CH_4}$, and $C_{CO_2}$ denote the concentration (vol%) and conversion of $CH_4$ and $CO_2$, respectively; and $W_{Ni}$ the nickel metal loading (wt%). Turnover frequencies (s$^{-1}$) were calculated from Eq. 5:

$$\text{TOF} = \frac{nCH_4}{\text{Weight of catalyst} \times W_{Ni} \times \frac{\text{Dispersion}}{58.7} \times 60} \qquad (5)$$

assuming a dispersion of 100% for 0.5Ni$_1$/HAP-Ce, and values calculated for 2Ni$_1$/HAP-Ce and 10Ni/HAP-Ce according to the following Eq. 6.

$$D(\%) = \frac{1.0092}{d_{V_A}} \qquad (6)$$

where $d_{V_A}$ represent the mean diameter of nickel particles (nm).

## Data availability

The data that support the findings of this study are available within the paper and its Supplementary Information, and all data are available from the authors on reasonable request.

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

## Acknowledgements

This paper is dedicated to the 70th anniversary of the Dalian Institute of Chemical Physics, Chinese Academy of Sciences. This work was supported by the National Key R&D Program of China (2016YFA0202801, 2017YFA0700104, and 2017YFA0402800); National Natural Science Foundation of China (21776270, 21606222, and 51761165012), Strategic Priority Research Program of the Chinese Academy of Sciences (XDB17020000), DNL Cooperation Fund, CAS (DNL180403) and LiaoNing Revitalization Talents Program (XLYC1807068). DFT calculations were supported by Beijing Natural Science Foundation (2184097). J.L. thanks National Program for Thousand Young Talents of China.

## Author contributions

M.A. performed the catalyst preparation, catalytic tests, and routine characterizations. S. Z. performed the DFT calculations. K.Z., W.X., and J.L. performed AC HAADF-STEM measurements. Y.S., X.P., X.L., and Lin.L. performed HRTEM measurements. A.F.L., M. A.I., and K.W. conducted in situ XPS measurements and data analysis. Y.R., Y.C., Lei.L., H.I., and Y.L. performed the EXAFS measurement and corresponding data analysis. W. W., Y.G., and Y.P. performed the MS detection. B.Q., A.W., X.W., and T.Z. designed the study, analyzed the data, and co-wrote the paper with input from A.F.L. All the authors discussed the results and commented on the manuscript and contributed to writing the STEM sections.

## Competing interests

The authors declare no competing interests.
