## [Peer Review File · Nature Communications]

Reviewers' comments:

Reviewer #1 (Remarks to the Author):

This manuscript reports the preparation of atomically dispersed Ni-based catalysts for the dry reforming of methane, which could convert the two typical green house gases, that is, methane and carbon dioxide, in to more useful feed stocks. The catalysts are well characterized, and computational method is used to simulate possible elementary reactions to help understand the catalytic performance. My recommendation is "acceptance after major revision". My concerns are the followings:

1. The coke-resistance of the catalyst, with atomically dispersed Ni single atoms stabilized by interaction with Ce-doped hydroxyapatite, is ascribed to its unique ability to only activate the first C-H bond in CH₄. This is ok. Under this consideration, the further cracking of the 2nd to 4th C-H bonds needs also to be addressed, since the major product obtained over these catalysts is syn-gas, although trace amount of ethylene is obtained.
2. According to the data reported in this manuscript, the obtaining of the catalyst with atomically dispersed Ni single atoms is realized by the very low loading of the Ni content. The authors are suggested to explain a little bite if it is possible to increase the metal loading of the catalyst while keeping the metal atom atomically dispersed.
3. According to the XRD results, the reduction of the 10 wt% Ni samples under H₂ at 500 °C results in the aggregation of Ni or simultaneous reduction of NiO and aggregation of Ni. What states should the nickel species be after preparation? The authors are suggested to explain this issue clearly in this paragraph to avoid possible confusion.
4. According to the XPS and TPR results of the 0.5Ni₁/HAP-Ce sample, its reduction at 500 °C could not give out metal Ni. Does this imply that Ni(OH)₂ or NiO is the active phase of the reaction? Furthermore, the authors correlated the characterization data of the catalysts reduced at 500 °C to the catalytic performance exhibited at 750 °C. Could the structure of the catalyst samples be kept under reaction stream from 500 °C to 750 °C? The authors are suggested to provide evidence to show what would happen or would not from 500 °C to 750 °C.
5. Concerning the different stability between 0.5Ni₁/HAP-Ce and 2Ni₁/HAP-Ce, the results show that the later one is more stable than the first. The authors explain this better stability of 2Ni₁/HAP-Ce also by the presence of atomically dispersed Ni. If so, 0.5Ni₁/HAP-Ce should be better because there is only atomically dispersed Ni (stable nickel species), while the nano-clusters and nano-particles on 2Ni₁/HAP-Ce (occupying about 30% according to Figure 1, less stable compared to atomically dispersed Ni) would make the catalyst be deactivated. The authors are suggested to discuss this issue in more detail.
6. The authors also suspect that Ni(OH)₂ might be the active phase of the reaction, if so, the authors are suggested to explain why 0.5Ni₁/HAP-Ce, 2Ni₁/HAP-Ce, and 10Ni₁/HAP-Ce show almost the same TOF, since the existence of nickel species on the three catalysts are very different. Of course, different nickel species might be active for the DRM reaction, they should not exhibit the same activity in terms of TOF.
7. The caption of Figure 3 should be improved to avoid possible confusion.

Reviewer #2 (Remarks to the Author):

Review of the manuscript NCOMMS-19-12378 entitled: "Atomically dispersed nickel as coke-resistant active sites for methane dry reforming"

The manuscript of Qiao, Zhang and co-workers discusses a currently very popular topic of single atom and single site Ni catalysts. The authors report on a coke resistant catalyst in the dry reforming of methane reaction, reasoned in the special nature of Ni single sites in activating methane. As matrices/supports for the active Ni species different types of hydroxyapatite are used. Based on DFT calculations they claim another type of DRM mechanism dealing with the selective C-H activation of

only one C-H bond per methane molecule and as a consequence a coke resistant catalyst. The manuscript is well organized and the argumentation is conclusive, at least based on the provided data. However, I do not see the level of novelty which is crucial to publish in Nature Communications. The Ni single atoms and cluster are not new, the stability of these systems are known (e.g. a reference not cited: J. Am. Chem. Soc., 2019, 141 (6), pp 2451–2461). In addition, the coke resistant character of isolated Ni cluster on MgO were already discussed (e.g. a reference also not cited: ACS Catal., 2018, 8 (10), pp 9821–9835). So basically the already published HAP approach was adapted to Ni and doped with Ce (J. Am. Chem. Soc., 2016, 138 (1), pp 56–59). As I mentioned above, the manuscript is interesting, but not suitable for Nature Comm.

There are several points I missed in the manuscript, three are mentioned exemplarily below:

- 1) No product analysis of the DRM reaction. CO? H₂? Other CH's? carbon balance? To show only the conversion of the educts, rate and TOF is not enough, in particular for a mechanistic suggestion!
- 2) The XPS results in Fig.2 b (peak splitting at ca. 855 eV) indicate not only the Ni(OH) and NiO contribution, it first of all gives an indication for Ni-O-Ni interaction, which is an evidence for not isolated Ni species (no SAC) see e.g. (Phys. Rev. B 54, 7716 – Published 15 September 1996).
- 3) The occurrence of Ni clusters (Ni₄MgO, see ACS reference) might also be the explanation for the coke resistance without an alternative, incomplete mechanistic adaption.

Reviewer #3 (Remarks to the Author):

A very interesting study. There are several key points that need clarification before I recommend acceptance in Nature Communications.

- 1) Methane activation and methane dry reforming have been studied MANY times on Ni/ceria systems. Configurations have been reported for the activation of methane at relatively low temperature and stable during the dry reforming without coke deposition. What is the NEW contribution of this article that justifies publication in a high level journal like Nature Communications?
- 2) The authors need to provide more clear evidence that they are really dealing with single atoms of Ni embedded on the oxide. Their TEM data is not conclusive.
- 3) The DFT calculations in Figure 5 do not give a good explanation of the lack of coke formation. Is the thermochemistry for C deposition and coke formation really uphill?

Response to reviewers

MS: NCOMMS-19-12378

Title: Atomically dispersed nickel as coke-resistant active sites for methane dry reforming

Author: Akri et al

We thank all the reviewers for their comments and have responded constructively to these below.

Reviewer #1:

1. The coke-resistance of the catalyst, with atomically dispersed Ni single atoms stabilized by interaction with Ce-doped hydroxyapatite, is ascribed to its unique ability to only activate the first C-H bond in CH₄. This is ok. Under this consideration, the further cracking of the 2nd to 4th C-H bonds needs also to be addressed, since the major product obtained over these catalysts is syn-gas, although trace amount of ethylene is obtained.

The reviewer raises an interesting question regarding activation (cleavage) of additional C-H bonds in reactively-formed methyl. For DRM, there are two possible reaction pathways following cleavage of the first C-H bond in methane and resulting generation of surface CH₃* species. Conventional wisdom holds that additional C-H bonds are sequentially broken to produce C and H₂, as illustrated in Scheme S1, and the surface C then oxidized by O* (from CO₂ dissociation) to form CO. This is the widely accepted DRM mechanism for supported Ni nanoparticle catalysts, and accounts for coking observed in nanoparticle systems due to incomplete oxidation of surface carbon. Alternatively, surface CH₃* may undergo direct oxidation by O* (from CO₂ dissociation) to CH₃O, which is then dehydrogenated stepwise to CO as recently proposed for Ni₄ single-site (*ACS Catal.* **2018**, 8, 9821) and Ni-Ru single-atom (*JACS* **2019**, 141, 7283) catalysts.

We have undertaken new DFT calculations to explore the second reaction pathway over our Ni₁/HAP-Ce catalyst, which reveal that the oxidation of CH₃ to CH₃O, and subsequent stepwise dehydrogenation to CO, is indeed energetically favorable, being overwhelmingly exothermic and with a modest activation barrier for each step. These additional calculations are shown in Figure 6b-c, and suggest that the rate-determining step for CO production is activation of the C-H bond of CH₃O (0.90 eV barrier) and not activation of the C-H bond in CH₃ (1.54 eV barrier). An expanded discussion of possible reaction pathways and DFT calculations is now included in the manuscript.

Figure 6. (b) Potential energy diagram from DFT calculations of CH₃ oxidation and CH₃O dehydrogenation, and (c) corresponding geometries over Ni₁/CeO₂. Numbers in parentheses indicate the activation barriers for elementary steps in eV. Optimized structures for reaction intermediates are shown inset (Ce: yellow, Ni: blue, O: red, C: black, H: white).

Action: Manuscript amended

2. According to the data reported in this manuscript, the obtaining of the catalyst with atomically dispersed Ni single atoms is realized by the very low loading of the Ni content. The authors are suggested to explain a little bite if it is possible to increase the metal loading of the catalyst while keeping the metal atom atomically dispersed.

The reviewer raises an excellent question in the context of optimizing performance for commercial application.

We would first like to highlight that a metal low loading is not necessarily a limitation, since our 0.5 wt% Ni₁/HAP-Ce is already sufficiently active to achieve complete CH₄ and CO₂ conversion. In our catalyst system, increasing the Ni loading to 2 wt% resulted in a small fraction of sub-nanoclusters and trace nanoparticles (Figure 1c), indicating an upper limit <2 wt% to maintain atomically dispersed Ni. To better quantify the maximum metal loading able to sustain single Ni atoms, we prepared a new 1 wt% Ni/HAP-Ce sample which contains no Ni nanoparticles and only trace 1 nm nanoclusters (Figure S5), indicating that almost all nickel was atomically dispersed; corresponding CO₂ and CH₄ conversions (Figure 4c and Figure S10b) for 1 wt% Ni/HAP-Ce were outstanding, and identical to that of the 2 wt% Ni/HAP-Ce. This additional data is included in the manuscript and supporting information and discussed in the text.

Note that an elegant recent study showed that CeO₂ nanorods stabilized 5 wt% single metal atoms (2.5 wt% Ni+2.5 wt% Ru, *JACS* **2019**, *141*, 7283), and such nanorods can also support high loadings of Au (*Nat. Commun.* **2016**, *7*, 13481) and Pt (*ACS Catal.* **2018**, *8*, 6203). Engineering the support properties (morphology and redox character) therefore offers a promising route to increase the loading of single metal atoms. Of course, achieving a high loading of single metal atoms is only relevant if they remain catalytically active, and hence a key feature of our work is demonstrating that not only can we stabilize single Ni atoms over Ce-doped HAP, but that these atoms exhibit an exceptional coke resistance. Furthermore, significant performance benefits can accrue for our catalytic system, even with loadings that do not exclusively give rise to single atoms: our 2 wt% Ni₁/HAP-Ce catalyst offers similar activity and superior stability for DRM (under identical conditions) to the 2.5 wt% Ni+2.5 wt% Ru/CeO₂ nanorods described above (Figure S11c), without recourse to precious metal. This additional reaction data is also described in the revised manuscript.

Figure S5. STEM images of 1Ni₁/HAP-Ce after reduction at 500 °C. Yellow arrows highlight nickel nanoclusters.

Figure S11c. (c) Stability of 2 wt% Ni₁/HAP-Ce during DRM at 600 °C. Reaction condition: 2 vol% CH₄ + 2 vol% CO₂ + 96 vol% He.

Action: Manuscript+ESI amended

3. According to the XRD results, the reduction of the 10 wt% Ni samples under H₂ at 500 °C results in the aggregation of Ni or simultaneous reduction of NiO and aggregation of Ni. What states should the nickel species be after preparation? The authors are suggested to explain this issue clearly in this paragraph to avoid possible confusion.

HAADF-STEM images of 10 wt% Ni samples after preparation but before H₂ reduction (Figure S7a-b) evidence small (~1-2 nm) NiO clusters/nanoparticles. This is now explicitly stated in the manuscript.

Figure S7. STEM images of (a) 10Ni/HAP and (b) 10Ni/HAP-Ce without reduction.

Action: Manuscript amended

4. According to the XPS and TPR results of the 0.5Ni₁/HAP-Ce sample, its reduction at 500 °C could not give out metal Ni. Does this imply that Ni(OH)₂ or NiO is the active phase of the reaction? Furthermore, the authors correlated the characterization data of the catalysts reduced at 500 °C to the catalytic performance exhibited at 750 °C. Could the structure of the catalyst samples be kept under reaction stream from 500 °C to 750 °C? The authors are suggested to provide evidence to show what would happen or would not from 500 °C to 750 °C.

XPS and TPR show that the chemical state of Ni atoms in 0.5Ni₁/HAP-Ce is consistent with Ni²⁺ species, however the local coordination environment will be very different from that in bulk NiO or Ni(OH)₂ due to the absence of Ni-O-Ni and Ni-(OH)-Ni bonds. Deactivation of 0.5Ni₁/HAP, 10Ni/HAP-Ce and 10Ni/HAP-Ce (Figure 4a and c) is accompanied by a loss of Ni(OH)_x surface species (Figure 3a, c-d), strongly evidencing Ni(OH)_x as the catalytically active phase. We have now conducted a 750 °C H₂ reduction of our 0.5Ni₁/HAP-Ce (and 2Ni₁/HAP-Ce) catalyst, and observed no performance deterioration compare to a 500 °C pre-reduced catalyst (Figure S11), confirming Ni remains sinter resistant and atomically dispersed. These additional results are discussed in the manuscript.

Figure S11. CO₂ and CH₄ conversion during DRM over (a) 0.5Ni₁/HAP and (b) 2Ni₁/HAP-Ce supported Ni catalysts reduced at 750 °C.

Action: Manuscript+ESI amended

- Concerning the different stability between 0.5Ni₁/HAP-Ce and 2Ni₁/HAP-Ce, the results show that the later one is more stable than the first. The authors explain this better stability of 2Ni₁/HAP-Ce also by the presence of atomically dispersed Ni. If so, 0.5Ni₁/HAP-Ce should be better because there is only atomically dispersed Ni (stable nickel species), while the nano-clusters and nano-particles on 2Ni₁/HAP-Ce (occupying about 30% according to Figure 1, less stable compared to atomically dispersed Ni) would make the catalyst be deactivated. The authors are suggested to discuss this issue in more detail.

In theory, all active sites within single-atom catalysts are identical and homogeneously distributed. However, in practice the fabrication of a completely homogeneous dispersion of single-atom sites is extremely difficult, and there is likely a distribution of coordination numbers and micro-environments (*JACS* **2017**, *139*, 10790). For our 0.5Ni₁/HAP-Ce we suggest that some single atoms present in lower coordination environments (e.g. the edges of HAP crystals) may aggregate during high temperature reaction to lower their surface energies, resulting in the (albeit very slow) deactivation observed in Figures S4c and S10b. In contrast, unstable Ni single atoms present in the as-prepared 2Ni₁/HAP-Ce likely sintered into the small clusters and nanoparticles observed following subsequent pre-reduction. Hence the reduced 2Ni₁/HAP-Ce catalyst may possess a smaller proportion of unstable single atoms than the reduced 0.5Ni₁/HAP-Ce catalyst.

Action: Manuscript amended

- The authors also suspect that Ni(OH)₂ might be the active phase of the reaction, if so, the authors are suggested to explain why 0.5Ni₁/HAP-Ce, 2Ni₁/HAP-Ce, and 10Ni/HAP-Ce show almost the same TOF, since the existence of nickel species on the three catalysts are very different. Of course, different nickel species might be active for the DRM reaction, they should not exhibit the same activity in terms of TOF.

The observation of a common TOF across a family of related catalysts is well-known in heterogeneous catalysis for diverse reactions (*ACS Catal.* **2015**, *5*, 6249; *Nano Res.* **2015**, *8*, 2913; *JACS* **2017**, *139*, 14150; *Angew. Chem. Int. Ed.* **2018**, *57*, 7795). There are several possible explanations for this. The simplest is that it reflects the existence of a common active site, present in different concentrations, whose reactivity dominates the entire catalytic performance. However, although 0.5Ni₁/HAP-Ce and 2Ni₁/HAP-Ce both contain a significant proportion of (highly active) single atoms which may dominate their reactivity, 10Ni/HAP-Ce predominantly comprises Ni clusters and nanoparticles. We therefore prefer an alternative hypothesis. CH₄ activation is generally regarded as rate-limiting for overall DRM (*J. Phys. Chem. B* **2004**, *108*, 4094; *Angew. Chem. Int. Ed.* **2016**, *55*, 7455), with a barrier around 0.9 eV over a Ni(111) surface representative of Ni nanoparticles in 2Ni₁/HAP-Ce, and 10Ni/HAP-Ce (*Angew. Chem. Int. Ed.* **2016**, *55*, 7455 and references therein). This barrier is very similar to that calculated for activation of the first C-H bond in CH₃O dehydrogenation over Ni single atoms in 0.5Ni₁/HAP-Ce. The similar TOFs for low and high loading Ni/HAP-Ce catalysts may therefore simply be a coincidence of similar energy barriers for different rate-determining steps. This aspect is now discussed in the manuscript.

Action: Manuscript amended

- The caption of Figure 3 should be improved to avoid possible confusion.

We Figure caption has been simplified.

Reviewer #2:

- “Ni single atoms and cluster are not new, the stability of these systems are known (e.g. a reference not cited: *J. Am. Chem. Soc.*, 2019, 141 (6), pp 2451–2461”

We wholeheartedly agree that reporting of Ni single atoms and clusters is not new. Indeed, our own group recently published two papers on Ni single atom catalysts (*Nature Energy* **2018**, 3, 140; *Angew. Chem. Int.-Ed.* **2018**, 57, 7071) for electrochemical CO₂ reduction and biomass hydrogenation. However, the article mentioned by the reviewer (*JACS* **2019**, 141, 2451) relates to the application of Ni single atom catalysts to CO₂ reduction instead of the dry reforming of CH₄ to syngas as in our present work, and hence has little relevance to its novelty.

The novelty of our work arises not only from the first report of Ni single atom catalysts for the dry reforming of CH₄ to syngas, but also from our demonstration that Ni single-atom sites are intrinsically coke resistant and accompanying (computationally underpinned) explanation. Although a second article mentioned by the reviewer (*ACS Catal.* **2018**, 8, 9821) does relate to dry methane reforming, the active species in that publication were Ni₄ clusters and not Ni single atoms; in fact, Ni single atoms were reported there as inactive for methane reforming. We show that Ni single atoms supported on hydroxyapatite are both active and intrinsically stable, the latter due to hitherto unreported selective C-H bond cleavage of methane. The discussion has been expanded throughout to highlight the superior activity, stability, and alternative reaction pathway, offered by single atom Ni/HAP-Ce catalysts for DRM.

Action: Manuscript amended

- “The coke resistant character of isolated Ni cluster on MgO were already discussed (e.g. a reference also not cited: *ACS Catal.*, 2018, 8 (10), pp 9821–9835”.

Coke resistance was indeed discussed in the referenced article (*ACS Catal.* **2018**, 8, 9821), however: (i) this resistance was (predicted) for Ni₄ clusters, not Ni single atoms which as noted above were reported as inactive for methane reforming in this publication; and (ii) was only theoretical and unsupported by any experimental evidence. Our work is therefore the first to (experimentally and theoretically) demonstrate that Ni single atoms are coke resistant in methane dry reforming, and furthermore show that this resistance is intrinsic to single atoms due to the inability to decompose CH₄ into carbon. We now cite the Ni₄/Mg(100) model catalyst study in our discussion on coke resistance.

Action: Manuscript amended

- “So basically the already published HAP approach was adapted to Ni and doped with Ce (*J. Am. Chem. Soc.*, 2016, 138 (1), pp 56–59)”

The article referenced by the reviewer above (*J. Am. Chem. Soc.* **2016**, 138, 59) was published by our group and describes “*Strong Metal–Support Interactions between Gold Nanoparticles and Nonoxides*”; it does not relate to either Ni or single atom catalysis, or to methane reforming (but rather CO and benzyl alcohol oxidation). To our knowledge, the present submission is the first to describe the creation of a single atom catalyst (involving any element) using a hydroxyapatite support, and hence there is significant novelty in the catalyst design in addition.

Action: No change

1. No product analysis of the DRM reaction. CO? H₂? Other CH's? carbon balance? To show only the conversion of the educts, rate and TOF is not enough, in particular for a mechanistic suggestion!

We apologize for this omission and now include discussion of the carbon balance and H₂/CO ratio (Figure S10c). Carbon balances for 0.5Ni₁/HAP-Ce and 2Ni₁/HAP-Ce catalysts were almost 100 % due to negligible carbon deposition, and the H₂/CO ratios were also stable around unity. However, corresponding values for the high loading 10Ni/HAP-Ce catalyst decreased gradually with reaction time, reflecting significant carbon deposition.

Figure S10. (c) Carbon balance and H₂/CO ratio over 0.5Ni₁/HAP-Ce, 2Ni₁/HAP-Ce, and 10Ni/HAP-Ce supported Ni catalysts during DRM.

Action: Manuscript+ESI amended

2. The XPS results in Fig.2 b (peak splitting at ca. 855 eV) indicate not only the Ni(OH) and NiO contribution, it first of all gives an indication for Ni-O-Ni interaction, which is an evidence for not isolated Ni species (no SAC) see e.g. (Phys. Rev. B 54, 7716 – Published 15 September 1996).

We thank the reviewer for bringing the work in Phys. Rev. B 1996, 54, 7716 to our attention, but note that Fig. 1 of that article shows nonlocal screening of Ni 2p core holes is only significant for NiO nanostructures with dimensions >10 ML. Assuming a NiO interplanar spacing of ~0.24 nm (Sci. Rep. 2015, 5, 14385), peak splitting due to such screening would only be observed for NiO nanoparticles >2.4 nm, of which there is no evidence in our 0.5Ni₁/HAP-Ce material by either HRTEM (Fig. 1 and S3-4) or XRD (Fig. S2). We can therefore be confident that the high binding energy peak in our Ni 2p XP spectra in Fig. 3b at 856.3 eV arises from isolated Ni atoms in a distinct Ni(OH) chemical environment, and not from Ni-O-Ni interactions in NiO nanoparticles. Additional discussion of the Ni 2p XP peak assignments is now included.

Action: Manuscript amended

3. The occurrence of Ni clusters (Ni₄MgO, see ACS reference) might also be the explanation for the coke resistance without an alternative, incomplete mechanistic adaption.

We acknowledge that Ni₄ clusters may also contribute to the coke resistance of our catalysts in DRM as reported in *ACS Catal.* **2018**, 8, 9821. However, we have no evidence for any such Ni₄ clusters from AC-HAADF-STEM imaging, and our DFT calculations confirm that DRM is energetically favorable over isolated Ni atoms without any need to invoke larger ensembles. However, we believe that Ni₁/HAP-Ce catalysts operate by a similar reaction mechanism to that proposed over Ni₄, as now discussed in the manuscript (Figure 6).

Action: Manuscript amended

Reviewer #3:

1. Methane activation and methane dry reforming have been studied MANY times on Ni/ceria systems. Configurations have been reported for the activation of methane at relatively low temperature and stable during the dry reforming without coke deposition. What is the NEW contribution of this article that justifies publication in a high level journal like Nature Communications?

We agree that ceria supported Ni catalysts have been widely studied in DRM reaction, albeit with the support basicity often invoked to explain resulting activity improvements and/or coke resistance (*ACS Sustain. Chem. Eng.* **2017**, 5, 2330; *Angew. Chem. Int. Ed.* **2017**, 56, 13041; *J. Phys. Chem. C* **2018**, 122, 11789; *Catal. Sci. Technol.* **2016**, 6, 851; *J. Phys. Chem. C* **2012**, 116, 10009; *Appl. Catal. B.* **2015**, 179, 128). However, there is only one prior report of total coke resistance (ref 14 in our manuscript, *Angew. Chem. Int. Ed.* **2016**, 55, 7455) under mild (450 °C) DRM conditions, which resulted in low catalytic activity; indeed the authors noted for their Ni/CeO₂ catalyst that “Once the first hydrogen is removed from the reactant molecule, a quick CH₃ → CH₂ → CH → C transformation occurs on the surface...”. This contrasts with our current study in which coke formation was completely suppressed, due to the operation of a different reaction pathway as revealed by additional DFT calculations (Figure 6).

Action: Manuscript amended

2. The authors need to provide more clear evidence that they are really dealing with single atoms of Ni embedded on the oxide. Their TEM data is not conclusive.

Our evidence for single Ni atoms is not derived solely from HAADF-STEM, although we believe that the aberration-corrected (AC-STEM) images in Figure 1 and Figures S3-4 shown below are extremely convincing.

Figures S3-4. AC HAADF-STEM images of 0.5Ni₁/HAP and 0.5Ni₁/HAP-Ce. Yellow circles highlight isolated nickel atoms.

We have now obtained additional direct evidence for Ni single atoms from X-ray adsorption spectroscopy (Figure 2 and Figure S8) of the reduced 2Ni₁/HAP-Ce catalyst, which is dominated by nearest-neighbour Ni-O and Ni-Ce shells, with only a very weak contribution from Ni-Ni scattering (Table S3). This confirms that Ni atoms are highly dispersed, coordinated to the HAP (or Ce-doped HAP) support and not present in nanoclusters or nanoparticles. XAS data are now discussed in the manuscript.

Table S3. Ni K-edge XAFS fitted parameters for 2Ni₁/HAP-Ce, 10Ni/HAP-Ce, and references.

Samples	Shell	N	R (Å)	$\sigma^2 \times 10^2$ (Å ²)	ΔE_0 (eV)	R-factor (%)
Ni Foil	Ni-Ni	12.0	2.48	0.6	4.5	0.001
NiO	Ni-O	6.0	2.08	0.4	12.8	0.02
	Ni-Ni	12.0	2.95	0.6	9.1	
10Ni/HAP-Ce	Ni-O	2.8	2.09	0.4	0.7	0.3
	Ni-Ni	6.8	2.50	0.6	-5	
2Ni ₁ /HAP-Ce	Ni-O	5.7	2.05	0.3	-2.9	1.9
	Ni-Ni	0.7	2.52	0.5	-8.6	
	Ni-Ce	3.9	2.60	0.5	-8.6	

Action: Manuscript+ESI amended

3. The DFT calculations in Figure 5 do not give a good explanation of the lack of coke formation. Is the thermochemistry for C deposition and coke formation really uphill?

As discussed in the response to Reviewer #1, our Ni single atom catalysts appear to avoid coke formation by directing DRM through a largely exothermic pathway involving dehydrogenation of a reactively-formed CH₃O intermediate. New DFT calculations reveal that the oxidation of CH₃ to CH₃O, and subsequent stepwise dehydrogenation to CO, is energetically favorable, being overwhelmingly exothermic and with a modest activation barrier for each step (Figure 6b-c). The rate-determining step for CO production is activation of the C-H bond of CH₃O (0.90 eV barrier) and not activation of the C-H bond in CH₃ (1.54 eV barrier), thereby avoiding a slightly endothermic pathway to coke formation (Figure 6a). These additional DFT calculations are now included in the manuscript.

Figure 6. (b) Potential energy diagram from DFT calculations of CH₃ oxidation and CH₃O dehydrogenation, and (c) corresponding geometries over Ni₁/CeO₂. Numbers in parentheses indicate the activation barriers for elementary steps in eV. Optimized structures for reaction intermediates are shown inset (Ce: yellow, Ni: blue, O: red, C: black, H: white).

Action: Manuscript amended

Reviewers' comments:

Reviewer #1 (Remarks to the Author):

The authors addressed most of the reviewers' comments. Now the manuscript is improved. However, there are still several key points that need to be clarified before I recommend acceptance.

1. The caption of Figure 1 should be changed according to the sequence: (a), (b), (c), (d), (e) instead of (a), (b), (d), (c), (e).

2. Please verify carefully the data presented in Figure 2(b). It seems that the text describing the data and the marked parts in the figure do not be in agreement. The Ni-O distance of 2.05\AA could not be clearly seen in the figure. How to distinguish the "Ni-Ni scattering distances at $\sim 2.50\text{\AA}$ " and the "Ni-Ce scattering distance of 2.60\AA "?

3. In Fig. 4 (b) and (d), please add the on stream time of the catalyst samples for DRM reaction before the TG experiment.

4. The authors ascribed the deactivation of 0.5Ni1/HAP sample to sintering of Ni species using TEM results to show the increase of Ni particles, I wonder why they did not provide XRD profiles to evidence the growth of Ni particles. The further discussion about the faster deactivation of 0.5Ni1/HAP-Ce versus 2Ni1/HAP-Ce also needs more considerations. In Figure 1, it is shown that nanoclusters and nano-particles of Ni did exist. One should answer why the nanoclusters and nano-particles did not aggregate in the reaction, while the more isolated single-atom did. However, the catalyst with nanoclusters and nano-particles exhibited long time stability, whereas the new 1 wt% Ni/HAP-Ce sample containing trace amount of nanoclusters with no Ni nanoparticles exhibited lower stability than 2Ni1/HAP-Ce (Figure 4(c)), why? Generally, in the preparation process, Ni would occupy the low energy site of the support firstly, then with the increase of Ni loading, Ni aggregates to form nano particles, thus Ni species on 0.5Ni1/HAP-Ce would be more stable.

Reviewer #2 (Remarks to the Author):

The authors answered all of my concerns and really improved the manuscript to a level acceptable to be published in Nature Communications.

Reviewer #3 (Remarks to the Author):

The response of the authors to my previous comments is satisfactory and I recommend acceptance of the manuscript for publication

Response to reviewers

MS No.: NCOMMS-19-12378A-Z

Title: Atomically dispersed nickel as coke-resistant active sites for methane dry reforming

Author: Akri et al

Reviewer #1 (Remarks to the Author):

1. The caption of Figure 1 should be changed according to the sequence: (a), (b), (c), (d), (e) instead of (a), (b), (d), (c), (e).

We thank the reviewer for their suggestion and have revised the figure caption and the order of appearance of figure panes 1c and 1d in the revised manuscript.

Figure 1. HAADF-STEM images of (a) 0.5Ni₁/HAP-Ce, (b) 2Ni₁/HAP-Ce and (c) 10Ni₁/HAP-Ce samples after 500 °C H₂ reduction. (d) Particle size distributions of (a), (b) and (c); yellow circles indicate atomically dispersed Ni and red squares indicate Ni metals nanoparticles. (e) EDX element maps of 0.5Ni₁/HAP-Ce

Action: Manuscript amended

2. Please verify carefully the data presented in Figure 2(b). It seems that the text describing the data and the marked parts in the figure do not be in agreement. The Ni-O distance of 2.05 Å could not be clearly seen in the figure. How to distinguish the “Ni-Ni scattering distances at ~2.50 Å” and the “Ni-Ce scattering distance of 2.60 Å”?

We apologize for any confusion between the data in Figure 2(b) and associated text. This apparent discrepancy arose from our presentation of spectra (which had not been phase shift corrected in the figure, as stated in the figure caption), while values in the text referred to scattering distances that had been phase shift corrected. We accept that it is clearer

for the reader to see phase shift corrected spectra, and have therefore amended the figure appropriately (as shown below) for consistency with the values in the text.

Ni-Ni scattering distances are readily identified by fitting to parameters derived from the Ni foil reference (first coordination shell at ~ 2.50 Å); such scattering pairs were necessary to obtain a good fit for the 10Ni/HAP-Ce catalyst. The inclusion of Ni-Ni scatters at ~ 2.5 Å was only just within the bounds of statistical significance for 2Ni₁/HAP-Ce, and the goodness of fit improved on fitting a Ni-Ce shell at 2.6 Å; this distance is much longer than any Ni-Ni bond. The accuracy of nearest neighbor metal-metal bond lengths for EXAFS data with a high signal:noise is typically better than ± 0.02 Å (*J. Phys. C: Solid State Phys.* **1987**, *20*, 4005; *J. Phys.: Conf. Ser.* **2009**, *190*, 012028).

Action: Manuscript amended

3. In Fig. 4 (b) and (d), please add the on stream time of the catalyst samples for DRM reaction before the TG experiment.

The on-stream reaction time has been added to panels (b) and (d) in a revised Figure 4 (see below).

Action: Manuscript amended

4. The authors ascribed the deactivation of 0.5Ni₁/HAP sample to sintering of Ni species using TEM results to show the increase of Ni particles, I wonder why they did not provide XRD profiles to evidence the growth of Ni particles. The further discussion about the faster deactivation of 0.5Ni₁/HAP-Ce versus 2Ni₁/HAP-Ce also needs more considerations. In Figure 1, it is shown that nanoclusters and nano-particles of Ni did exist. One should answer why the nanoclusters and nano-particles did not aggregate in the reaction, while the more isolated single-atom did. However, the catalyst with nanoclusters and nano-particles exhibited long time stability, whereas the new 1 wt% Ni/HAP-Ce sample containing trace amount of nanoclusters with no Ni nanoparticles exhibited lower stability than 2Ni₁/HAP-Ce (Figure 4(c)), why? Generally, in the preparation process, Ni would occupy the low energy site of the support firstly, then with

the increase of Ni loading, Ni aggregates to form nanoparticles, thus Ni species on 0.5Ni₁/HAP-Ce would be more stable.

The reviewer makes an excellent suggestion that XRD (in addition to TEM) should evidence on-stream sintering of Ni species in 0.5Ni₁/HAP if this is the cause of deactivation. XRD analysis of the post-reaction sample confirms the emergence of reflections characteristic of fcc Ni metal, and hence that deactivation is (at least partially) associated with sintering of Ni species and nanoparticle formation. This evidence is highlighted in a revised Figure S2d (see below) and described in the manuscript.

Action: Manuscript+ESI amended

Regarding the second question, 1Ni₁/HAP-Ce and 2Ni₁/HAP-Ce catalysts exhibit almost identical stabilities for over 60 h on-stream, see Figure 4(c) below; it is unclear why the reviewer believes that the 1Ni₁/HAP-Ce deactivates faster.

We would like to clarify that nanoclusters and nanoparticles did in fact sinter during reaction, albeit with slower kinetics than that of isolated atoms (reflecting the higher surface energy and hence instability of the latter). We agree that single atoms are expected to preferentially occupy low energy sites on the support, and hence exhibit higher

(relative) stability than randomly distributed single atoms. However, even if all isolated Ni atoms initially occupy low energy surface sites, provided that the strength of their interaction with the HAP-Ce support is lower than the cohesive energy between Ni atoms, then thermodynamics will always drive sintering for a sufficiently high energy input, e.g. at high temperature. A comment to this effect is now included in the discussion

Action: Manuscript amended

REVIEWERS' COMMENTS:

Reviewer #1 (Remarks to the Author):

The authors have addressed my concerns, and I propose acceptance of the manuscript.